# Characteristics of Patients with Dry Eye Who Switched from Long-Acting Ophthalmic Solution to Diquafosol Ophthalmic Solution

**DOI:** 10.3390/jcm14082790

**Published:** 2025-04-18

**Authors:** Sho Ishikawa, Takafumi Maruyama, Koichiro Murayama, Kei Shinoda

**Affiliations:** 1Department of Ophthalmology, Saitama Medical University, Saitama 350-0495, Japanshinok@saitama-med.ac.jp (K.S.); 2Tsuruse Murayama Eye Clinic, Saitama 354-0021, Japan

**Keywords:** diquafosol ophthalmic solution, dry eye, long-acting, polyvinylpyrrolidone, P2Y2 receptor agonist

## Abstract

**Background/Objectives:** Long-acting (extended) diquafosol ophthalmic solution 3% (DQSLX) is administered less frequently (three times daily) than diquafosol ophthalmic solution (DQS) (six times daily). However, some patients do not prefer DQSLX because of perceived stickiness. We investigated the subjective and objective characteristics of patients with dry eye who switched from using DQSLX to DQS. **Methods:** We retrospectively enrolled 51 patients (11 men and 40 women) whose eye drop prescription was changed from DQSLX to DQS between June 2024 and September 2024. Subjective symptoms, tear break-up time, and fluorescein-staining scores were evaluated from baseline to 4 weeks after DQS use. We asked the participants to choose between DQS and DQSLX 4 weeks after using DQS. **Results:** In total, 51 eyes of 51 patients (11 men and 40 women; mean age: 68.2 ± 14.7 years) were enrolled. The DQS group showed significant worsening of the subjective symptoms, tear break-up time, and fluorescein staining scores (20.8 ± 22.5, 5.2 ± 3.4, and 1.6 ± 2.0, respectively) relative to the baseline (15.9 ± 18.7, 6.3 ± 3.2, and 0.7 ± 1.4, respectively) (*p* = 0.003, *p* < 0.001, and <0.001, respectively). Eleven (21.6%) patients expressed their preference for continuing DQS because of the good sensation of the eye drops. An analysis of the group that preferred the DQS ophthalmic solution revealed no significant changes in subjective symptoms or fluorescein staining scores after DQS treatment. **Conclusions:** DQSLX improved the subjective symptoms and objective findings of patients with dry eye relative to DQS.

## 1. Introduction

Dry eye disease is a complex, multifactorial condition characterized by an unstable tear film, leading to ocular discomfort and potential visual impairment while adversely affecting the ocular surface [1,2]. Dry Eye Workshop II outlines a series of diagnostic assessments for dry eye disease, including subjective symptom evaluation, tear break-up time (TBUT), tear osmolarity, and ocular surface staining. Additionally, factors such as lipid abnormalities, meibomian gland dysfunction, and tear volume are assessed to classify subtypes [1]. Dry eye disease is broadly categorized into aqueous-deficient and evaporative types. Therapeutic strategies for managing tear-deficiency dry eye include artificial tears, anti-inflammatory agents, secretagogues, and tear retention therapies [3]. Conversely, the Asia Dry Eye Society classifies dry eye into three categories: aqueous deficiency, increased evaporation, and decreased wettability [2]. Decreased wettability occurs when the corneal surface fails to retain water, despite normal tear volume and evaporation rates. Membrane mucins contribute to the corneal epithelium’s water-wetting properties, and their reduced expression has been observed in cases of dry eye with decreased wettability [2,4].

Diquafosol ophthalmic solution (DQS) is a purinergic P2Y2 receptor agonist approved in Japan for the treatment of dry eye disease. This 3% ophthalmic solution promotes the secretion of both tear fluid and mucins [5,6]. Randomized, double-blind, multicenter clinical trials have demonstrated that DQS leads to significantly greater improvements in fluorescein and rose bengal staining scores compared to a placebo. Additionally, DQS has been shown to be non-inferior to sodium hyaluronate ophthalmic solution 0.1% in improving fluorescein staining scores and more effective than sodium hyaluronate 0.1% in enhancing rose bengal staining scores [7,8,9,10]. Long-term studies have further confirmed the sustained efficacy of DQS, with improvements noted in both subjective dry eye symptoms and real-world clinical settings [5]. Moreover, DQS 3% has been effective in treating specific subtypes of dry eye, including aqueous-deficient dry eye, short tear film break-up time dry eye, and obstructive meibomian gland dysfunction [5,6,7,8,9,10]. Despite its efficacy, poor adherence to the recommended dosing schedule remains an issue. The package insert advises administration six times daily; however, a study by Uchino et al. reported that only 8.3% of patients adhered to this regimen [11]. Previous studies suggest that patients who consistently use the prescribed dosage experience greater symptom improvement compared to those who use drops only as needed [11,12].

A long-acting formulation of diquafosol ophthalmic solution 3% (DQSLX) has also been approved in Japan. DQSLX incorporates polyvinylpyrrolidone (PVP) as an additive, which extends its duration of action. PVP has been reported to interact electrostatically with secretory and membrane mucins, as well as water molecules [13,14]. The concurrent action of DQS, which enhances mucin secretion, and PVP, which binds to mucins and promotes tear fluid retention, is believed to prolong the therapeutic effect. Unlike the standard formulation requiring administration six times per day, DQSLX necessitates only three daily applications, potentially improving patient compliance. It reportedly improves corneal and conjunctival staining scores relative to placebo [13]. Two studies have reported that DQSLX improves subjective symptoms and corneal and conjunctival staining scores relative to DQS [15,16]. The better symptoms and objective findings of the DQSLX group than those of the DQS group were attributed to the non-adherence of several participants in the DQS group to the prescribed frequency of eye drop administration [16].

However, no studies have compared the subjective symptoms and objective findings of patients using DQSLX and those using DQS when administered according to the package insert instructions. This study aimed to evaluate whether switching from DQSLX to standard DQS, when used according to prescribing recommendations, affects subjective symptoms and objective dry eye parameters.

## 2. Materials and Methods

### 2.1. Participants

Institutional Review Board/Ethics Committee approval was obtained from the Ethics Committee of Saitama Medical University Hospital (hospital 2024-106). This study adhered to the principles of the Declaration of Helsinki. Due to its retrospective design, the requirement for written informed consent was waived by the Ethics Committee of Saitama Medical University.

A retrospective analysis was conducted on 70 patients whose prescriptions were switched from DQSLX to DQS at Saitama Medical University Hospital or Tsuruse Murayama Eye Clinic between June and September 2024. Exclusion criteria included patients using alternative dry eye treatments, those who had used DQSLX for less than three months, individuals using over-the-counter dry eye medications or lubricating drops, and those taking oral medications to enhance tear fluid production (e.g., pilocarpine). Additional exclusions applied to those who used DQS fewer than six times daily as prescribed; had diabetes, ocular infections, glaucoma, or allergic conjunctivitis requiring eye drop treatment; underwent ophthalmic surgery within one month; had a history of punctal plug use or surgical interventions for dry eye; or did not provide consent.

The primary assessment focused on subjective dry eye symptoms documented in medical records. Both institutions utilized the Symptom Assessment in Dry Eye (SANDE) questionnaire, a validated tool employing a 100 mm visual analog scale to quantify the severity and frequency of dry eye symptoms, including dryness and irritation [17]. The SANDE score ranges from 0 to 100, with the average values used for statistical analysis.

Furthermore, data regarding the frequency of eye drop administration were collected. Participants were questioned about their DQS or DQSLX instillation frequency over the past month, selecting from predefined options (0, 1, 2, 3, 4, 5, 6, 7, or 8 times per day). Patients who did not adhere to the prescribed regimen—six times daily for DQS and three times daily for DQSLX—were excluded.

Objective evaluations included dry eye parameters, fluorescein staining scores, and TBUT. Vital staining was performed using 2 µL of a preservative-free 1% fluorescein dye solution, instilled into the conjunctival sac via micropipette. Fluorescein staining was graded on a 0–9 scale [18,19]. TBUT was measured with fluorescein solution without anesthesia, and participants were instructed to blink several times to ensure dye distribution. The time from the last blink to the first appearance of a corneal black spot in the stained tear film was recorded three times, with the average value used for analysis.

Comparisons of subjective symptoms and objective findings were conducted before and four weeks after transitioning to DQS. Patients were then asked whether they preferred to continue with DQS or revert to DQSLX, and their reasons were extracted from medical records. Additionally, adverse effects associated with DQS were evaluated during the study period.

### 2.2. Statistical Analyses

All statistical analyses were performed using JMP version 17 (SAS Institute, Tokyo, Japan). All data are expressed as means ± standard deviation. The Wilcoxon signed-rank test was used to compare the SANDE, fluorescein staining, and TBUT scores before and after the 4-week administration of DQS. An unpaired t-test was used to compare the ages of the groups preferring DQS and DQSLX. Fisher’s exact test was used to compare the sexes of the groups preferring DQS and DQSLX. The Mann–Whitney U test was used to compare the SANDE score, fluorescein staining score, and TBUT of the groups preferring DQS and DQSLX. All analyses of objective findings used values from the right eye. Statistical significance was set at *p* < 0.05.

## 3. Results

Due to non-compliance with the prescribed frequency of DQS eye drops, 19 patients (27%) were excluded from the study. Consequently, the final analysis included 51 eyes from 51 patients (11 men and 40 women) with an average age of 68.2 ± 14.7 years. No patients with aqueous-deficient dry eye had Sjögren’s syndrome.

At week 4 after commencement, the study group showed a significant worsening of symptoms (SANDE score), the fluorescein staining score, and TBUT (20.8 ± 22.5, 1.6 ± 2.0, and 5.2 ± 3.4, respectively) compared with the baseline (15.9 ± 18.7, 0.7 ± 1.4, and 6.3 ± 3.2, *p* = 0.003, *p* < 0.001, and <0.001, respectively) (Table 1, Appendix A). At 4 weeks after using DQS, 40 participants preferred to resume DQSLX eye drops, while 11 preferred to continue DQS eye drops. The participants were divided into two groups based on their preference for DQSLX or DQS eye drops, and their characteristics were analyzed. There were no significant differences between the two groups in terms of age, sex, subjective symptom score, staining score, or TBUT (*p* = 0.51, 0.42, 0.15, 0.34, and <0.06, respectively) (Table 2). In the preferred DQS group, no patient revealed fluorescein corneal staining.

At week 4 of treatment, those who preferred DQSLX showed a significant worsening of the symptom score, fluorescein staining score, and TBUT (24.3 ± 23.5, 1.8 ± 2.0, and 4.8 ± 3.2, respectively) relative to the baseline (18.3 ± 19.9, 0.9 ± 1.5, and 6.0 ± 3.2; *p* = 0.004, *p* < 0.001, and <0.001, respectively) (Table 3). For the preferred DQS group, TBUT was significantly worse after 4 weeks (6.5 ± 3.7) of treatment relative to the baseline (7.2 ± 3.2) (*p* = 0.008), while the symptom and fluorescein staining scores did not significantly change at 4 weeks (8.2 ± 12.5 and 0.9 ± 1.3) from the baseline (7.3 ± 10.1 and 0 ± 0; *p* = 0.52 and 0.20, respectively) (Table 4).

The reasons for the preferences of the participants were evaluated. Among those who preferred DQSLX eye drops, 20 participants provided valid responses (50%). The most common reasons were a preference for fewer instillations (*n* = 14), increased dryness after switching to DQS eye drops (*n* = 4), and greater eye fatigue associated with DQS eye drops (*n* = 2). Among those who preferred DQS eye drops, seven participants provided valid responses (64%). The primary reasons included the sticky sensation of DQSLX eye drops (*n* = 3), increased ocular discharge with DQSLX eye drops (*n* = 2), a film-like sensation over the eyes with DQSLX eye drops (*n* = 1), and excessive tearing with DQSLX eye drops (*n* = 1).

Adverse events were assessed during the observation period after switching from DQSLX to DQS. Mild instillation discomfort was reported in two (3.9%) patients; however, no participant discontinued the treatment due to this sensation. Additionally, none of the patients reported increased ocular discharge or conjunctival hyperemia.

## 4. Discussion

In our study, we demonstrated that changing from DQSLX eye drops to DQS eye drops worsened the subjective dry eye symptom score (SANDE score) and other objective findings (the fluorescein corneal staining score and TBUT) of dry eye, even when the number of eye drops used was as the package insert states. Eleven (22%) patients opted to continue DQS eye drops because of unpleasant feelings or adverse drug reactions to the DQSLX eye drops, increased ocular discharge, a film-like sensation, and excessive tearing. Forty (78%) patients opted to return to using DQSLX eye drops because of their effectiveness, fewer instillations, decreased dryness, and improved eye fatigue. At week 4 after switching from DQSLX eye drops to DQS eye drops, the DQSLX group showed a significant worsening of the symptom score, fluorescein staining score, and TBUT relative to baseline, whereas no significant changes were observed for the DQS group. The adverse drug reactions included instillation eye discomfort (3.9%) at 4 weeks after switching from DQSLX to DQS.

The incorporation of PVP distinguishes DQSLX from DQS. Originally introduced in the 1950s as a blood plasma expander for trauma patients, PVP has since been utilized in various pharmaceutical formulations, including tablets and liquid preparations [20]. It serves as an additive in ophthalmic solutions and contact lens lubricants, with studies suggesting that its inclusion enhances comfort during contact lens wear [21]. PVP interacts electrostatically with secretory and membrane mucins, as well as water molecules [13,14]. DQS is known to stimulate water secretion from conjunctival epithelial cells and mucin release from conjunctival goblet cells via P2Y2 receptor activation [22]. The extended efficacy of DQSLX is thought to result from the combined actions of DQS and PVP, wherein DQS enhances mucin secretion while PVP–mucin binding facilitates the prolonged retention of tear fluid on the ocular surface, thereby extending the therapeutic effect.

DQS is a uridine 5′-triphosphate (UTP) derivative and a potent P2Y2 receptor agonist. P2Y2 receptors are found on the palpebral and bulbar conjunctival epithelial cells and on the corneal epithelium and meibomian gland cells, where they are involved in the regulation of aqueous fluid and mucin secretion. By stimulating P2Y2 receptors, DQS promotes tear film stabilization [23]. DQS eye drops improved dry eye symptoms, and fluorescein and rose bengal staining scores were lower with DQS than with placebo [7,8] and sodium hyaluronate ophthalmic solution 0.1% [9,10] in randomized, double-blind, multicenter trials. A phase 3 study evaluating DQSLX demonstrated comparable efficacy and safety to placebo in patients with dry eye disease. Significant improvements in fluorescein corneal and lissamine green conjunctival staining scores were observed at weeks 2 and 4 compared to placebo. However, TBUT did not show a significant improvement with DQSLX relative to placebo [13]. In the present study, symptom scores, TBUT, and fluorescein staining scores exhibited greater improvement in the DQSLX group than in the DQS group at week 4. These findings are consistent with those of a meta-analysis of randomized clinical trials comparing DQS with placebo [10]. In this study, a significant difference in TBUT was observed, which differed from the results of a phase 3 trial of DQSLX ophthalmic solution [13]. Previous studies comparing DQS ophthalmic solution (used for 6 weeks) [7] and DQSLX ophthalmic solution (used for 4 weeks) [12] with placebo reported that short-term use does not result in a significant improvement in TBUT; however, long-term use leads to a significant improvement in TBUT [9,24]. Furthermore, comparisons of sodium hyaluronate ophthalmic solutions and DQS revealed a significant difference in TBUT, even with short-term use [8]. These findings suggest that the activation of P2Y2 receptors promotes the secretion of secretory mucin when starting from an untreated state of dry eye. This stabilizes the ocular surface, which affects TBUT after a prolonged duration. However, the ocular surface may be stabilized to some extent in patients who have already been treated with DQS ophthalmic solution because of the aqueous layer and mucin secretion, leading to a more rapid change in TBUT. In this study, patients who had been using DQSLX ophthalmic solution for more than 3 months were selected, suggesting that a significant change in TBUT was observed when switching from DQSLX to DQS ophthalmic solution.

Two studies investigated changes in the ocular surface following a switch from a DQS ophthalmic solution to a DQSLX ophthalmic solution [15,16]. In a study by Kaido et al., evaluations conducted 4 weeks after switching to DQSLX eye drops demonstrated significant improvements in subjective symptoms, TBUT, and corneal staining scores compared with those observed with the use of DQS eye drops [15]. Similarly, we observed significant improvements in subjective symptoms (SANDE scores) and corneal staining scores 4 weeks after switching from DQS to DQSLX eye drops in our study. However, no significant differences were observed in the TBUT [16]. In our previous report, we attributed the superior improvement in dry eye symptoms and clinical findings with DQSLX compared with DQS eye drops to poor adherence to the DQS regimen. Specifically, only 5.6% of patients adhered to the prescribed six daily instillations of DQS eye drops, whereas 88.9% adhered to the three daily instillations of DQSLX eye drops. This significant difference in adherence suggests that the insufficient dosing of DQS eye drops may have prevented their intended therapeutic effects from being fully realized. In dry eye disease, patients who use eye drops less frequently than the dosage specified in the package insert exhibit worse subjective symptom scores relative to those who adhere to the prescribed dosing regimen [11,12]. Therefore, patients who did not adhere to the prescribed dosing frequencies were excluded. Despite this exclusion, our results were consistent with those of previous reports [15,16], demonstrating that DQSLX eye drops were significantly superior to DQS eye drops in terms of subjective symptoms, TBUT, and corneal staining scores. These findings suggest that DQSLX eye drops may provide greater therapeutic benefits for dry eye disease relative to DQS eye drops. DQSLX suggestively increases mucin levels through the P2Y2 receptor, and the binding of PVP to this secreted mucin prolongs the duration of action, as well as the action of mucin on the ocular surface. Therefore, DQSLX may be more effective than DQS.

We analyzed the data by dividing the participants into two groups: those who preferred DQSLX eye drops and those who preferred DQS eye drops. In the group that preferred DQSLX eye drops, switching to DQS eye drops resulted in a deterioration of subjective symptom scores, TBUT, and corneal staining scores relative to the baseline, which was consistent with the results obtained from the overall patient analysis. In contrast, there was no significant difference in subjective symptom scores or corneal staining scores for the group that preferred DQS eye drops after switching to DQS eye drops compared with the baseline when they were using DQSLX eye drops. There are two possible explanations for this finding. The first is the perception of symptom changes. The group that preferred DQSLX eye drops may have experienced worsening symptoms after switching to DQS eye drops, leading to a preference for DQSLX eye drops. In contrast, the group that preferred DQS eye drops may not have perceived a significant worsening of symptoms after switching and did not feel the need to return to their previously used DQSLX eye drops. Kaido et al. reported changes in subjective symptoms, TBUT, and corneal staining scores 4 weeks after switching from DQS to DQSLX eye drops. In their study, a subgroup analysis was conducted by dividing the participants into two groups based on their corneal vital staining scores: those with a score of 0 and those with a score of 1 or higher. The results showed that switching to DQSLX eye drops led to improvements in subjective symptoms, TBUT, and corneal vital staining scores for the group with a corneal vital staining score of 1 or higher. In contrast, there were no significant differences in subjective symptom scores, TBUT, or corneal vital staining scores for the group with a corneal vital staining score of 0 relative to the baseline using DQS eye drops [15]. A possible explanation for these findings is that switching to DQSLX eye drops does not result in noticeable changes in cases of mild dry eye, as DQS eye drops provide sufficient therapeutic effects. Similarly, the group that preferred DQS eye drops had a corneal staining score of 0 in our study. They also had lower subjective symptom scores and longer TBUTs than the DQSLX eye drop group, although the differences were statistically insignificant. This suggests that the group that preferred DQS had relatively mild cases. Therefore, it is speculated that several patients in this group did not feel the need to return to using DQSLX eye drops because DQS eye drops had already provided sufficient improvement in dry eye symptoms. The second reason pertains to the instillation experience and side effects of eye drops. In this study, we conducted a survey on the reasons for continuing with the selected eye drops. The reasons cited by the group that preferred to continue using DQSLX eye drops included fewer instillations, increased dryness after switching to DQS eye drops, and greater eye fatigue associated with DQS eye drops. In contrast, the reasons for preferring to continue DQS eye drops included the sticky sensation of DQSLX eye drops, increased ocular discharge with DQSLX eye drops, a film-like sensation over the eyes with DQSLX eye drops, and excessive tearing with DQSLX eye drops. These findings suggested that the primary reason for returning to DQSLX eye drops in the DQSLX-preferred group was the perceived reduction in effectiveness after switching to DQS eye drops. In contrast, the DQS-preferred group continued using DQS eye drops mainly because of the discomfort and side effects associated with DQSLX eye drops. The group that wished to continue DQS eye drops likely found them sufficiently effective while being concerned about the adverse effects and poor instillation experience of DQSLX eye drops. Conversely, the group that preferred to continue DQSLX eye drops may have done so because DQS eye drops were insufficient to control dry eye symptoms or because they preferred a lower instillation frequency. These results suggest that while DQSLX eye drops may be more effective than DQS eye drops in improving both subjective dry eye symptoms and objective findings, they may have a less favorable instillation experience and a higher likelihood of side effects than DQS eye drops.

The main adverse effects of diquafosol ophthalmic solution include increased eye discharge, stickiness, and eye irritation, with reported frequencies ranging from 1.8% to 10.6% [13,15,16]. The incidence of adverse effects was similar for the DQS and DQSLX ophthalmic solutions [13]. However, one study reported that 3.6% to 10.6% of patients experienced increased eye discharge when switching from DQS to DQSLX [15,16]. In the present study, no increased eye discharge was observed when switching from DQSLX to DQS. The increase in eye discharge can be attributed to mucin secretion induced by diquafosol through P2Y_2_ receptor activation, and this is corroborated by its transparent appearance. As the mucin secreted by DQSLX is retained for a prolonged period because of PVP, it is presumed that eye discharge may occur in patients switching from DQS to DQSLX, even if they previously had no symptoms. Additionally, two cases of eye irritation were observed in the present study. PVP reportedly reduces ocular discomfort and irritation [21]. It is speculated that switching to DQS leads to the disappearance of the protective effects of PVP, resulting in the onset of irritation symptoms.

This study has some limitations. First, the sample size was small, which may have caused bias. In this study, we recruited patients using only DQS and excluded those using multiple eye solutions. Furthermore, patients who were unable to use the eye drops according to the prescribed frequency described in the package insert were excluded. As a result, patients with severe dry eye may have been inadvertently excluded, potentially influencing adherence rates. To mitigate bias, data were collected from both primary and tertiary hospitals; however, further research is required to evaluate the impact of disease severity on adherence to different eye drop administration frequencies. Additionally, the study duration for DQS treatment was relatively short. Since adherence was monitored for only four weeks, potential bias may exist. Previous research by Uchino reported that the proportion of patients adhering to the prescribed frequency for 1–3 months after initiating treatment was low (DQS: 5%, sodium hyaluronate ophthalmic solution: 3%, and rebamipide ophthalmic suspension: 14%) [11]. If the observation period extends beyond 1 month, adherence to the prescribed dosing regimen may further decline. Since dry eye disease is a chronic condition, additional studies of its long-term use are warranted. Finally, the patients may not have accurately reported their dosing frequency. In this study, patients were asked to report the number of daily instillations using a questionnaire, and their usual dosing frequency was confirmed through interviews. In addition, they were asked whether they could maintain this frequency on a daily basis. However, it was not possible to verify whether the reported dosing frequency was consistently followed, given the retrospective nature of this study. Consequently, the exact number of daily instillations remains uncertain. If patients did not reach the dosing frequency specified in the package insert, the intended therapeutic effects of the eye drops may not have been fully realized, potentially introducing bias into the study results.

## 5. Conclusions

We demonstrated that DQS eye drops were less effective than DQSLX eye drops in improving subjective symptom scores, TBUT, and corneal staining scores when patients adhered to the prescribed dosing frequency specified in the package insert. Patients who preferred DQS eye drops tended to have milder dry eye disease, as indicated by a corneal staining score of zero and relatively mild subjective symptoms, than those who preferred DQSLX eye drops. Additionally, these patients tended to prefer DQS eye drops over DQSLX eye drops due to adverse effects and the sticky sensation associated with DQSLX eye drops.

## Figures and Tables

**Table 1 jcm-14-02790-t001:** Profile of ocular surface and outcomes of using preferred eye drops.

	Baseline	After 4 Weeks	*p*-Value
Symptoms score	15.9 ± 18.7	20.8 ± 22.5	0.003 *
Fluorescein staining score	0.7 ± 1.4	1.6 ± 2.0	<0.001 *
TBUT (s)	6.3 ± 3.2	5.2 ± 3.4	<0.001 *
Preferred eye drops (LX/DQS)		40/11	

* Wilcoxon signed-rank test. TBUT: tear break-up time.

**Table 2 jcm-14-02790-t002:** Profiles of subgroups before switching to DQS.

	Preferred DQSLX Group (*n* = 40)	Preferred DQS Group (*n* = 11)	*p*-Value
Age (years)	67.5 ± 15.5	70.8 ± 11.2	0.51 *
Sex (female/male)	30/10	10/1	0.42 †
Symptoms score	18.3 ± 19.9	7.3 ± 10.1	0.15 ‡
Fluorescein staining score	0.9 ± 1.5	0 ± 0	0.34 ‡
TBUT (s)	6.0 ± 3.2	7.2 ± 3.2	0.06 ‡

* unpaired *t*-test. † Fisher’s extract test. ‡ Mann–Whitney U test. DQS: diquafosol ophthalmic solution. DQSLX: long-acting diquafosol ophthalmic solution. TBUT: tear break-up time.

**Table 3 jcm-14-02790-t003:** Dry eye parameter values for group preferring DQSLX.

Preferred DQSLX (*n* = 40)	Baseline	After 4 Weeks	*p*-Value
Symptoms score	18.3 ± 19.9	24.3 ± 23.5	0.004 *
Fluorescein staining score	0.9 ± 1.5	1.8 ± 2.0	<0.001 *
TBUT (s)	6.0 ± 3.2	4.8 ± 3.2	<0.001 *

* Wilcoxon signed-rank test. DQSLX: long-acting diquafosol ophthalmic solution. TBUT: tear break-up time.

**Table 4 jcm-14-02790-t004:** Dry eye parameter values for group preferring DQS.

Preferred DQS (*n* = 11)	Baseline	After 4 Weeks	*p*-Value
Symptoms score	7.3 ± 10.1	8.2 ± 12.5	0.52 *
Fluorescein staining score	0 ± 0	0.9 ± 1.3	0.20 *
TBUT (s)	7.2 ± 3.2	6.5 ± 3.7	0.008 *

* Wilcoxon signed-rank test. DQS: diquafosol ophthalmic solution. TBUT: tear break-up time.

## Data Availability

The original contributions presented in this study are included in the Appendix A. Further inquiries can be directed to the corresponding author.

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
