# Peer review of "Characteristics of Patients with Dry Eye Who Switched from Long-Acting Ophthalmic Solution to Diquafosol Ophthalmic Solution"

_jcm, 2025, doi:10.3390/jcm14082790_

Round 1
Reviewer 1 Report
Comments and Suggestions for Authors
I would like to congratulate the authors on this well written manuscript
It is detailed, clear, easy to follow
The introduction can be improved by making it more focused towards the study and title. Authors give a good overview of DED and treatments
It would be helpful if authors include a flow chart including how many patients in each arm and the treatment to make the methods easier to follow
Supplmental file has make and female symbols in additon to some notes, written in Japanese characters - would suggest switiching to english
Author Response
The introduction can be improved by making it more focused towards the study and title. Authors give a good overview of DED and treatments
Response 1 -Thank you for your suggestion. We rephased from “In the present study, we retrospectively compared the changes in eye drop use, TBUT, and fluorescein staining scores for patients who switched from DQSLX to DQS with adherence to eye drop frequency.” to “This study aimed to evaluate whether switching from DQSLX to standard DQS, when used according to prescribing recommendations, affects subjective symptoms and objective dry eye parameters.” (line 79-81, page 2)
It would be helpful if authors include a flow chart including how many patients in each arm and the treatment to make the methods easier to follow
Response 2-Thank you for your suggestion. Although the sentence “Due to non-compliance with the prescribed frequency of DQS eye drops, 19 patients (27%) were excluded from the study.” was added to emphasize the difficulty of adhering to 6 instillations per day, it made the explanation slightly more complex. Therefore, to avoid confusion as much as possible, we have placed this statement in the Results section rather than in the Methods. The study design follows the same methodology as that used in references 15 and 16.
Supplmental file has make and female symbols in additon to some notes, written in Japanese characters - would suggest switiching to english
Response 3-Thank you for your suggestion. We changed from Japanese characters to English (supplemental file).
Reviewer 2 Report
Comments and Suggestions for Authors
This manuscript addresses a clinically relevant question regarding the patient experience and treatment efficacy of two formulations of diquafosol ophthalmic solution—standard DQS and the long-acting DQSLX. The study is timely given the widespread use of secretagogues in dry eye management and the need to balance efficacy with patient adherence and comfort. The authors’ effort to compare subjective and objective outcomes retrospectively among patients switching between formulations is commendable. However, several points merit clarification and improvement to enhance scientific rigor and readability.
The retrospective nature of the study and strict adherence criteria are acknowledged strengths. However, the exclusion of 27% of patients due to non-compliance highlights the real-world challenge of maintaining adherence to frequent dosing regimens
The worsening of SANDE score, TBUT, and fluorescein staining after switching to DQS is well presented, but it is not entirely clear whether this reflects the inferiority of DQS or the superior retention/mucin affinity properties
Add a clear research aim, e.g., “This study aimed to evaluate whether switching from DQSLX to standard DQS, when used according to prescribing recommendations, affects subjective symptoms and objective dry eye parameters.
Author Response
The retrospective nature of the study and strict adherence criteria are acknowledged strengths. However, the exclusion of 27% of patients due to non-compliance highlights the real-world challenge of maintaining adherence to frequent dosing regimens
Response 1-Thank you for your suggestion. We changed from Japanese characters to English characters (supplemental file). There have been reports indicating that patients who do not adhere to the prescribed number of instillations tend to have worse subjective symptoms compared to those who do (References 11,12). Similarly, we also find that studies involving medications requiring frequent instillation are challenging to conduct. In this study, the exclusion of such patients resulted in a reduced sample size; therefore, we added the following sentence to the limitations paragraph in the Discussion section: "Furthermore, patients who were unable to use the eye drops according to the prescribed frequency described in the package insert were excluded."(line 341-342, page 8)
The worsening of SANDE score, TBUT, and fluorescein staining after switching to DQS is well presented, but it is not entirely clear whether this reflects the inferiority of DQS or the superior retention/mucin affinity properties
Response 2-Thank you for your suggestion. PVP interacts electrostatically with secretory and membrane mucins as well as water molecules (References 13 and 14). According to the Asia Dry Eye Society (Reference 2), a reduction in mucin leads to decreased wettability of the ocular surface, resulting in shortened TBUT and worsening of dry eye symptoms. Therefore, we hypothesize that, due to the absence of added PVP, DQS may retain mucin for a shorter duration compared to DQSLX, potentially leading to worse outcomes in terms of TBUT, staining scores, and subjective symptoms.
Add a clear research aim, e.g., “This study aimed to evaluate whether switching from DQSLX to standard DQS, when used according to prescribing recommendations, affects subjective symptoms and objective dry eye parameters.
Response 3-Thank you for your suggestion. We rephased from “In the present study, we retrospectively compared the changes in eye drop use, TBUT, and fluorescein staining scores for patients who switched from DQSLX to DQS with adherence to eye drop frequency.” to “This study aimed to evaluate whether switching from DQSLX to standard DQS, when used according to prescribing recommendations, affects subjective symptoms and objective dry eye parameters.”(line 79-81, page 2)